# ParaFlow: Parallel Sampling for Flow Matching Models

## Abstract

This paper approaches the fundamental challenge of accelerating the inherently autoregressive nature of sampling in Flow Matching (FM) models like Stable Diffusion 3 and Flux through a numerical systems perspective. Specifically, we introduce a unified framework that recasts the autoregressive sampling process as solving a system of triangular nonlinear equations (TNEs), thereby facilitating a paradigm shift toward non-autoregressive sampling featuring parallel vector field computation across multiple timesteps. Within this generic framework, we establish that: (1) the TNE system admits a unique solution corresponding precisely to the autoregressive sampling trajectory; (2) solving the TNE system guarantees convergence to this exact trajectory in far fewer sequential iterations. Building on these insights, we present *ParaFlow*, a training-free, step-parallel sampler for accelerating autoregressive FM samplers. Extensive experiments validate that ParaFlow achieves up to a $4\times$ reduction in sequential sampling steps and significant wall-clock speedup of up to $4.3\times$, with negligible impact on FID and CLIP scores. The source code will be released publicly.

## 1 Introduction

Flow Matching (FM) has emerged as a leading framework for high-fidelity generative modeling, powering recent state-of-the-art systems such as Stable Diffusion 3 (Esser et al., 2024) and Flux (Black Forest Labs, 2024). FM models learn a time-dependent vector field that transports a simple prior distribution to the data distribution via an ordinary differential equation (ODE). Thus, sampling reduces to numerically integrating this ODE, an inherently autoregressive procedure that requires tens of sequential evaluations of a large neural network, constituting a major bottleneck for real-time and interactive use.

Two main strategies have been explored to mitigate this cost. The first develops more efficient numerical solvers: high-order methods such as DPM-Solver (Lu et al., 2022a;b) and task-specific solvers (Shaul et al., 2023; Frankel et al., 2025) reduce the number of function evaluations (NFEs), enabling generation in fewer steps, but remain fully sequential. The second pursues model distillation (Salimans & Ho, 2022), compressing multi-step teachers into student models capable of few-shot or even one-shot generation. Approaches such as Latent Consistency Models (LCMs) (Luo et al., 2023), based on Consistency Models (Song et al., 2023), achieve strong results but require expensive retraining and degrade fidelity. Both methods reduce the step count but do not remove sequential dependencies.

In contrast to these approaches, our work departs from the sequential paradigm. We propose **ParaFlow**, a framework that accelerates sampling not by reducing steps but by executing them in parallel. Our key idea is to recast the entire sampling trajectory—ordinarily produced by an ODE solver—as a system of Triangular Nonlinear Equations (TNEs). This reformulation decouples dependencies across timesteps, enabling simultaneous evaluation of the vector field at multiple steps.

We show that the proposed TNE system admits a unique solution mathematically identical to the trajectory of the original autoregressive sampler, ensuring no loss in sample quality. To solve this system efficiently, we adopt a fixed-point iteration scheme that converges rapidly to the exact solution. A sliding-window implementation further makes the approach computationally feasible for long sampling chains.

Our main contributions are:

- We are the first to formulate the sampling process of Flow Matching models as a system of TNEs, enabling step-level parallelism.

- We propose an efficient, parallel fixed-point solver, ParaFlow, with theoretical guarantees of convergence to the exact autoregressive trajectory.

- Through extensive experiments on state-of-the-art flow-matching based models like Stable Diffusion 3 and Flux, we demonstrate wall-clock speedups of **1.4-4.3**× and a reduction in sequential operations by up to **4**×, with negligible impact on generation quality.

ParaFlow is a training-free, plug-and-play accelerator that is compatible with existing pre-trained FM models, opening a new direction for efficient model inference.

## 2    RELATED WORK

Accelerating generative model sampling is a long-standing research problem. For diffusion and flow-based models, which rely on iterative refinement, this is particularly critical.

**Numerical ODE Solvers for Flow Matching.** Flow Matching (FM) (Lipman et al., 2022; Liu et al., 2022; Chen & Lipman, 2024) frames the sampling process as the integration of a learned time-dependent vector field. Early work (Song & Ermon, 2020; Song et al., 2021) in score-based SDEs and their corresponding probability-flow ODEs enabled deterministic sampling and efficient likelihood evaluation. FM further refines this by training continuous normalizing flows (Chen et al., 2018) through conditional vector field regression along optimized probability paths, such as those (e.g. Lipman et al., 2022; Tong et al., 2023; Lipman et al., 2024; Schusterbauer et al., 2025; Wang et al., 2025) derived from diffusion or optimal transport, improving stability and speed with standard ODE solvers.

The choice of numerical integrator is crucial. First-order methods, such as Euler and DDIM (Song et al., 2020), are computationally inexpensive but require a large number of steps to achieve high-quality results. To mitigate this problem, second-order solvers, such as Heun's method, are frequently employed (Karras et al., 2022). For the most demanding tasks, higher-order solvers tailored specifically for diffusion models, such as the DPM-Solver family (Lu et al., 2022a;b; Zheng et al., 2023) and Bespoke Solvers (Shaul et al., 2023; 2024), offer improved sample quality with lower NFEs. Furthermore, methods like S4S (Frankel et al., 2025), AYS (Sabour et al., 2024) and DMN (Xue et al., 2024) have demonstrated potential for reducing discretization and global error. However, these methods still require strictly sequential execution, limiting opportunities for parallelization.

**Parallel Sampling.** ParaDiGMS (Shih et al., 2024) accelerates sampling by leveraging parallelized Picard iterations to simultaneously predict and iteratively refine future denoising steps. ParaTAA (Tang et al., 2024) reformulates the autoregressive diffusion sampling process as TNEs and uses fixed-point methods to solve multiple steps in parallel. ParaSolver (Lu et al., 2025) further models the sampling process as a system of banded nonlinear equations, and shows how to exploit structural properties and hierarchical initialization to preserve sample quality while greatly reducing inference time. Our proposed method, ParaFlow, builds on the parallel solving framework pioneered by those works, but is the first to **(i)** systematically formulate parallel sampling in the context of Flow Matching (FM) models; **(ii)** provide accompanying theoretical guarantees specific to FM-based parallel solving; and **(iii)** empirically validate that TNE-based parallel sampling works for state-of-the-art FM models without degradation in sample fidelity.

## 3    PRELIMINARY: FLOW MATCHING MODELS

Flow Matching models aim to learn a continuous-time process that transports a simple prior distribution $p_0$ (e.g., $\mathcal{N}(0, I)$) into a complex target data distribution $p_1$. This transformation is described by a time-dependent vector field $v_t : \mathbb{R}^d \times [0, 1] \rightarrow \mathbb{R}^d$, parameterized by a neural network with parameters $\theta$. For a sample trajectory $x_t, t \in [0, 1]$, the dynamics are governed by the ODE:

$$\frac{dx_t}{dt} = v_t(x_t, \theta), \quad x_0 \sim p_0. \tag{1}$$

The vector field $v_t$ is learned by optimizing its underlying neural network parameters, such that the distribution of the terminal state $x_1$ approximates the target distribution $p_1$ when $x_0 \sim p_0$.

**Sampling Process.** To generate new samples, one first draws an initial point $x_0 \sim p_0$ and then integrates the ODE in Eq. (1) from $t = 0$ to $t = 1$. In practice, this integration is carried out numerically. A common approach is the first-order Euler method: given a discretization $0 = t_0 < t_1 < \cdots < t_N = 1$, the update rule is

$$x_{t_{i+1}} = x_{t_i} + (t_{i+1} - t_i) \cdot v(x_{t_i}, t_i, \theta), \quad i = 0, \ldots, N - 1. \tag{2}$$

This discretization produces a sequence of intermediate states $x_{t_i}$. Since each update depends on the result of the previous step, the sampling procedure is inherently sequential, requiring $N$ successive evaluations of the neural network.

## 4 PROPOSED METHOD: PARAFLOW

### 4.1 MOTIVATION

The autoregressive update in Eq. (2) constitutes the main bottleneck: each step depends on the previous one, precluding parallel computation and limiting sampling efficiency. We formalize this dependency as follows.

**Definition 1** (Autoregressive Sampling Procedure). *Initiating with a sample $x_{t_0} \sim p_0$, the sampling process for an FM model using a numerical ODE solver is an autoregressive procedure of the form:*

$$x_{t_i} = x_{t_0} + \sum_{j=0}^{i-1} h_j \cdot v_j(x_{t_j}, t_j, \theta), \quad i \in \{1, \ldots, N\}, \tag{3}$$

*where $h_j = t_{j+1} - t_j$ is the step size and $v_j$ is the vector field evaluated at time $t_j$.*

This formulation highlights the stepwise dependency. Viewing the entire trajectory $\{x_{t_0}, x_{t_1}, \ldots, x_{t_N}\}$ as unknown variables, this autoregressive process can be cast as a system of $N+1$ equations. Solving this system yields the full trajectory simultaneously, eliminating sequential dependencies and enabling parallel computation.

### 4.2 RECASTING AUTOREGRESSIVE SAMPLING AS TRIANGULAR NONLINEAR EQUATION SOLVING

We can view the sequence of updates in Definition 1 as a system of TNEs. Let $\{\hat{x}_{t_0}, \ldots, \hat{x}_{t_N}\}$ be the unknown variables corresponding to the true sampling trajectory $\{x_{t_0}, \ldots, x_{t_N}\}$.

**Definition 2** (TNEs for Flow Matching Sampling). *We define the system of TNEs for the autoregressive sampling procedure as:*

$$\mathcal{F}(\hat{x}_{t_0}, \ldots, \hat{x}_{t_N}) = \begin{cases} \hat{x}_{t_0} - x_{t_0} = 0, \\ \hat{x}_{t_i} - F_{i-1}(\hat{x}_{t_0}, \ldots, \hat{x}_{t_{i-1}}) = 0, \quad i \in \{1, \ldots, N\}, \end{cases} \tag{4}$$

*where $x_{t_0}$ is the initial sample from the prior and $F_{i-1}$ is defined based on the ODE solver:*

$$F_{i-1}(\hat{x}_{t_0}, \ldots, \hat{x}_{t_{i-1}}) = \hat{x}_{t_0} + \sum_{j=0}^{i-1} h_j \cdot v(\hat{x}_{t_j}, t_j, \theta). \tag{5}$$

This reformulation decouples the stepwise dependencies: given tentative states $\{\hat{x}_{t_j}\}$, the vector field $v(\hat{x}_{t_j}, t_j, \theta)$ can be evaluated for all timesteps $j \in \{0, \ldots, N - 1\}$ in parallel. A crucial question then arises: does solving this system recover the same trajectory as the original sequential process?

**Proposition 1** (Trajectory Equivalence [see *App. B* for proof]). *The TNE system in* Eq. (4) *possesses a unique root that is identical to the sampling trajectory $\{x_{t_i}\}_{i=0}^{N}$ generated by the autoregressive procedure in* Eq. (3).

This proposition ensures that by solving the TNEs, we can produce a sample indistinguishable in quality from that obtained via the standard autoregressive process.

---

**Algorithm 1:** ParaFlow: Parallel Sampling within a Sliding Window

---

**Input** : Initial noise $x_{t_0}$, total steps $N$, timesteps $\{t_i\}_{i=0}^N$, tolerance $\delta$, window size $p$.

**Output** : Final sample $\hat{x}_{t_N}^{(K)}$.

1 Initialize $\{\hat{x}_{t_i}^{(0)} = x_{t_0}, i = 0, \ldots, p-1\}$.

2 $i \leftarrow 0, k \leftarrow 0;$

3 **while** $i < N$ **do**

4     Compute vector field $v_j^{(k)} = v(\hat{x}_{t_{i+j}}^{(k)}, t_{i+j}, \theta)$ in parallel for $j \in \{0, \ldots, p-1\}$.

5     Update temporary states $F_j \leftarrow \hat{x}_{t_i}^{(k)} + \sum_{l=0}^j (t_{i+l+1} - t_{i+l}) v_l^{(k)}$ in parallel for $j \in \{0, \ldots, p-1\}$.

6     Update the state $\hat{x}_{t_{i+j+1}}^{(k+1)} \leftarrow F_j$ in parallel for $j \in \{0, \ldots, p-1\}$.

7     *stride* $\leftarrow$ number of converged states based on $\frac{1}{D}\|\hat{x}_{t_{i+j+1}}^{(k+1)} - \hat{x}_{t_{i+j+1}}^{(k)}\|^2 \leqslant \delta^2$ for $j \in \{0, \ldots, p-1\}$.

8     Initialize new states outside current window: $\hat{x}_{t_{i+p+j}}^{(k+1)} \leftarrow \hat{x}_{t_{i+p}}^{(k+1)}$ for $j \in \{1, \ldots, stride\}$.

9     $i \leftarrow i + stride, \quad k \leftarrow k + 1, \quad p \leftarrow \min(p, N - i)$.

10 **Return:** $\hat{x}_{t_N}^{(K)}$

---

### 4.3 SOLVING THE TNE SYSTEM WITH FIXED-POINT ITERATION

We can solve the system in Eq. (4) using various root-finding methods. Standard fixed-point iteration (FPI) is a natural choice. Given an initial guess for the entire trajectory $\{\hat{x}_{t_i}^{(0)}\}_{i=0}^N$, the update rule is:

$$\hat{x}_{t_i}^{(k+1)} = F_{i-1}(\hat{x}_{t_0}^{(k)}, \ldots, \hat{x}_{t_{i-1}}^{(k)}), \quad i \in \{1, \ldots, N\}, \tag{6}$$

with $\hat{x}_{t_0}^{(k+1)} = x_{t_0}$ for all $k$. Crucially, this update is step-level parallelizable: for any iteration $k$, the vector field evaluations $v(\hat{x}_{t_j}^{(k)}, t_j, \theta)$ across all $j \in \{0, \ldots, N-1\}$ can be computed simultaneously.

**Proposition 2** (Convergence Guarantee [see *App. C* for proof])**.** *Starting from any initial guess $\{\hat{x}_{t_i}^{(0)}\}_{i=0}^N$, the fixed-point iteration in* Eq. (6) *converges exactly to the autoregressive sampling trajectory $\{x_{t_i}\}_{i=0}^N$. This convergence is achieved in at most $N$ iterations.*

The proof follows from the structure of the update mapping, which admits the autoregressive trajectory as its unique fixed point. In practice, the number of parallel iterations $K$ required for convergence is typically much smaller than $N$, resulting in substantial acceleration.

### 4.4 EFFICIENT IMPLEMENTATION WITH A SLIDING WINDOW

Solving for the entire trajectory of $N$ steps in parallel can be computationally demanding. To make this practical, we implement the FPI within a sliding window of size $p \ll N$. We perform parallel iterations only on $p$ subequations at a time. The window slides forward once the states within it have converged.

**Initialization.** We initialize all states within the first window to the starting noise vector: $\hat{x}_{t_i}^{(0)} = x_{t_0}$ for $i = 0, \ldots, p-1$. When the window slides, new states entering the window are initialized with the last converged state from the previous window.

**Stopping Criterion.** We define convergence within the window using a tolerance threshold $\delta$. An iterate $\hat{x}_{t_i}^{(k)}$ is considered converged if the change from the previous iteration is small: $\frac{1}{D}\|\hat{x}_{t_i}^{(k)} - \hat{x}_{t_i}^{(k-1)}\|^2 \leqslant \delta^2$, where $D$ is the number of pixels. The window slides forward by a stride, which is the number of contiguously converged states from the beginning of the window.

## 5 EXPERIMENT

**Experimental setup.** All experiments were carried out on 8 Ascend 910B GPUs. We evaluated generation quality using the Fréchet Inception Distance (FID) (Heusel et al., 2017) and CLIP score (Hessel et al., 2021), calculated over 10,000 random prompts from the FluxPrompting dataset (VincentGOURBIN, 2024). We report the average wall-clock time to generate a single

| Steps | Method | Stable Diffusion 3 | | | | | |
|---|---|---|---|---|---|---|---|
| | | Iters↓ | NFE↓ | CLIP↑ | FID↓ | Time (s)↓ | Speedup↑ |
| 100 | Euler | 100.00 | 100.00 | 32.38 | 50.75 | 18.22 | 1.0× |
| | Euler + ParaFlow | **26.47** | 211.78 | 32.37 | 51.62 | **5.55** | **3.3×** |
| 75 | Euler | 75.00 | 75.00 | 32.37 | 51.14 | 13.57 | 1.0× |
| | Euler + ParaFlow | **22.96** | 183.66 | 32.37 | 51.91 | **4.95** | **2.7×** |
| 50 | Euler | 50.00 | 50.00 | 32.37 | 51.57 | 9.37 | 1.0× |
| | Euler + ParaFlow | **19.94** | 159.49 | 32.36 | 51.95 | **4.43** | **2.1×** |
| 25 | Euler | 25.00 | 25.00 | 32.31 | 53.34 | 5.17 | 1.0× |
| | Euler + ParaFlow | **16.02** | 128.12 | 32.32 | 54.00 | **3.73** | **1.4×** |

Table 1: Quantitative comparisons of different methods on Stable Diffusion 3 over $10,000$ random samples at $1024 \times 1024$. The visual comparisons are shown in the Appendix. The best results in each step are highlighted in **bold**. "↑" (resp. "↓") means the larger (resp. smaller), the better. Note that Euler + ParaFlow is evaluated with tolerance 0.005.

image. Our baseline is the standard sequential Euler solver. We compare it with our proposed Euler + ParaFlow method across different total sampling steps ($N \in \{100, 75, 50, 25\}$). For ParaFlow, we use a default parallel window size of $p = 8$.

We define the total steps ($N$) as the number of discrete steps in the sampling schedule. For a standard sequential method such as the Euler solver, the number of iterations (Iters) and the total number of function evaluations (NFE) are identical to $N$. For our parallel method, ParaFlow, Iters signifies the reduced number of sequential blocks executed, while the total computational workload, NFE, reflects the parallel execution and is calculated as Iters $\times p$, where $p$ is the parallel window size.

## 5.1 Results on Stable Diffusion 3

We first evaluate ParaFlow on Stable Diffusion 3, generating $1024 \times 1024$ images from $10,000$ random prompts.

**Performance across different step counts.** Table 1 compares the results of the vanilla Euler solver and Euler + ParaFlow under different numbers of sampling steps. Our method demonstrates a significant reduction in latency by parallelizing the computation. For example, in the 100-step setting, ParaFlow reduces the number of sequential iterations from 100 to just 26.47, translating into a $3.3\times$ reduction in wall clock time (from 18.22 to 5.55 s), while incurring only a marginal increase in FID (50.75 to 51.62) and maintaining an identical CLIP score. The corresponding qualitative results are shown in Figure 1, further confirming the visual fidelity of our method.

This highlights the core trade-off of ParaFlow: we exchange a higher total computational cost (NFE from 100.00 to 211.78) for a substantial decrease in generation latency. This advantage holds even at fewer sampling steps, where ParaFlow delivers a $1.4\times$ speedup in the 25-step setting, underscoring its robustness.

**Effect of tolerance.** To further investigate the trade-off between speed and fidelity, we vary ParaFlow's tolerance parameter at 25 total steps (Table 2). Increasing the tolerance (e.g., 0.01) reduces the average number of iterations to 12.16, resulting in a $1.7\times$ acceleration, at the cost of a modest increase in FID (55.59). Conversely, employing a stricter tolerance (e.g., 0.005) yields a FID of 54.00, closely aligning with the baseline Euler method (53.34), while still achieving a $1.4\times$ speedup. This tunable parameter enables practitioners to adapt ParaFlow to task-specific requirements, providing a controllable trade-off between sample quality and inference efficiency. Representative visual comparisons are provided in Figure 2, which clearly illustrate the perceptual impact of different tolerance settings.

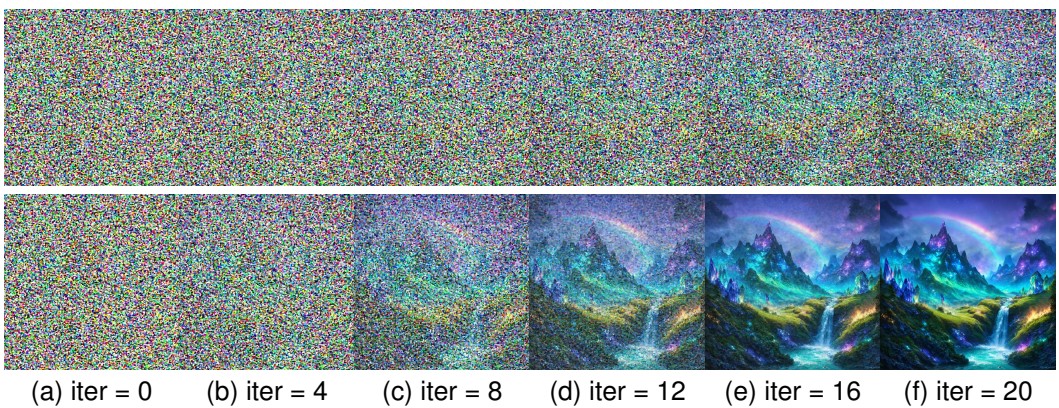

(a) iter = 0      (b) iter = 4      (c) iter = 8      (d) iter = 12      (e) iter = 16      (f) iter = 20

Figure 1: All images are generated by the **stable-diffusion-3-medium** model with 50 sampling steps, using the prompt *"Create a dreamlike landscape featuring rolling hills, shimmering waterfalls, and towering mountains made entirely of crystal and gemstones that refract light into rainbows."* at resolution $1024 \times 1024$. The **top row** corresponds to the **Euler** method, while the **bottom row** corresponds to our proposed **Euler + ParaFlow** method (tolerance = 0.005).

| Method | Tolerance | Stable Diffusion 3 | | | | | |
|---|---|---|---|---|---|---|---|
| | | Iters↓ | NFE↓ | CLIP↑ | FID↓ | Time (s)↓ | Speedup↑ |
| Euler | 0 | 25.00 | 25.00 | 32.31 | 53.34 | 5.17 | 1.0× |
| Euler + ParaFlow | 0.01 | **12.16** | 97.32 | 32.31 | 55.59 | **3.08** | **1.7×** |
| | 0.007 | 13.96 | 111.72 | 32.31 | 54.63 | 3.38 | 1.5× |
| | 0.005 | 16.02 | 128.12 | 32.32 | 54.00 | 3.73 | 1.4× |

Table 2: Quantitative comparisons on Stable Diffusion 3 with 25 sampling steps over $10,000$ random samples at $1024 \times 1024$. The best results are highlighted in **bold**. "↑" (resp. "↓") means the larger (resp. smaller), the better.

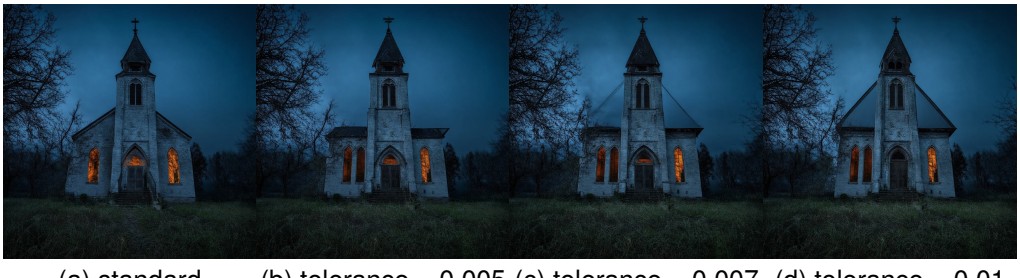

(a) standard      (b) tolerance = 0.005   (c) tolerance = 0.007   (d) tolerance = 0.01

Figure 2: The above images show results from the **stable-diffusion-3-medium** model at resolution $1024 \times 1024$ with 25 sampling steps. The prompt used is *"Abandoned church in night"*. The leftmost image corresponds to the **Euler** method, while the other three images correspond to our proposed **Euler + ParaFlow** method with parallel window size = 8 under different tolerance values (0.005, 0.007, 0.01).

## 5.2 RESULTS ON FLUX

We further validate ParaFlow on the Flux model, examining its performance across various sampling steps, resolutions, and window sizes.

| Steps | Method | Flux Model | | | | | |
|---|---|---|---|---|---|---|---|
| | | Iters↓ | NFE↓ | CLIP↑ | FID↓ | Time (s)↓ | Speedup↑ |
| 100 | Euler | 100.00 | 100.00 | 32.31 | 55.62 | 87.94 | 1.0× |
| | Euler + ParaFlow | **23.49** | 187.95 | 32.40 | 55.28 | **22.16** | **4.0×** |
| 75 | Euler | 75.00 | 75.00 | 32.36 | 55.25 | 66.15 | 1.0× |
| | Euler + ParaFlow | **19.32** | 154.54 | 32.42 | 55.35 | **18.38** | **3.6×** |
| 50 | Euler | 50.00 | 50.00 | 32.38 | 54.89 | 44.38 | 1.0× |
| | Euler + ParaFlow | **15.11** | 120.86 | 32.45 | 55.84 | **14.57** | **3.0×** |
| 25 | Euler | 25.00 | 25.00 | 32.40 | 55.23 | 22.60 | 1.0× |
| | Euler + ParaFlow | **12.79** | 102.33 | 32.47 | 55.88 | **12.42** | **1.8×** |

Table 3: Quantitative comparisons of different methods on Flux over $10,000$ random samples at $1024 \times 1024$. The visual comparisons are shown in the Appendix. The best results in each step are highlighted in **bold**. "↑" (resp. "↓") means the larger (resp. smaller), the better. Note that Euler + ParaFlow is evaluated with tolerance $0.01$.

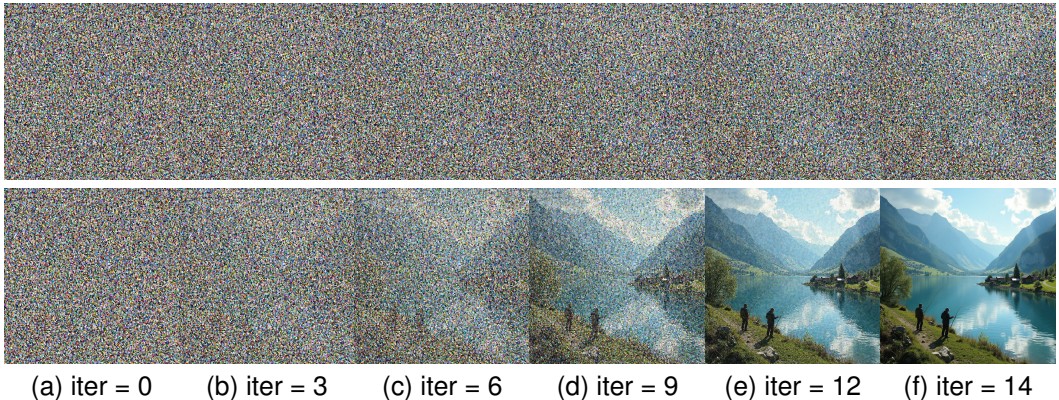

(a) iter = 0     (b) iter = 3     (c) iter = 6     (d) iter = 9     (e) iter = 12     (f) iter = 14

Figure 3: All images are generated by the **FLUX.1-dev** model with 50 sampling steps, using the prompt *"A peaceful mountain village surrounded by rolling hills and a sparkling lake in the foreground, with fishermen casting their lines into the calm waters."* at resolution $1024 \times 1024$. The **top row** corresponds to the **Euler** method, while the **bottom row** corresponds to our proposed **Euler + ParaFlow** method (parallel window size = 8, tolerance = 0.01).

**Performance across different steps.** As shown in Table 3, ParaFlow delivers even greater acceleration on Flux. At 100 steps, it reduces the required iterations to 23.49, yielding a $4.0\times$ speedup (87.94 vs. 22.16 s) while slightly improving the FID score. The acceleration remains consistent across step counts, reaching $1.8\times$ even at 25 steps, confirming ParaFlow's effectiveness as a general-purpose accelerator. Figure 3 further corroborates these findings with qualitative examples, highlighting that the perceptual quality is well preserved despite the substantial reduction in computation.

**Effect of resolution.** Table 4 demonstrates ParaFlow's scalability across different image resolutions with a 25-step schedule. Our method delivers consistent speedups: $2.3\times$ at $512 \times 512$, $1.8\times$ at $1024 \times 1024$, and $1.7\times$ at $2048 \times 2048$. Crucially, the impact on FID and CLIP scores remains minimal across all resolutions, proving that ParaFlow's benefits generalize effectively to both low- and high-resolution synthesis. Complementary visual results are provided in Figure 4, which illustrate that image fidelity is preserved across scales while the generation time is substantially reduced.

| Resolution | Method | Flux Model | | | | | |
|---|---|---|---|---|---|---|---|
| | | Iters↓ | NFE↓ | CLIP↑ | FID↓ | Time (s)↓ | Speedup↑ |
| 512×512 | Euler | 25.00 | 25.00 | 32.19 | 52.49 | 11.31 | 1.0× |
| | Euler + ParaFlow | **13.24** | 105.90 | 32.23 | 52.59 | **4.81** | **2.3×** |
| 1024×1024 | Euler | 25.00 | 25.00 | 32.40 | 55.23 | 22.60 | 1.0× |
| | Euler + ParaFlow | **12.79** | 102.33 | 32.47 | 55.88 | **12.42** | **1.8×** |
| 2048×2048 | Euler | 25.00 | 25.00 | 31.34 | 55.98 | 109.55 | 1.0× |
| | Euler + ParaFlow | **14.30** | 114.37 | 31.31 | 56.46 | **65.86** | **1.7×** |

Table 4: Quantitative comparisons of Euler and Euler + ParaFlow on the Flux model over $10,000$ random samples at 25 sampling steps. The best results in each resolution are highlighted in **bold** for Iters, Time, and Speedup only. "↑" (resp. "↓") means the larger (resp. smaller), the better. Note that Euler + ParaFlow is evaluated with tolerance $0.01$ for $512 \times 512$ and $1024 \times 1024$, and $0.0028$ for $2048 \times 2048$.

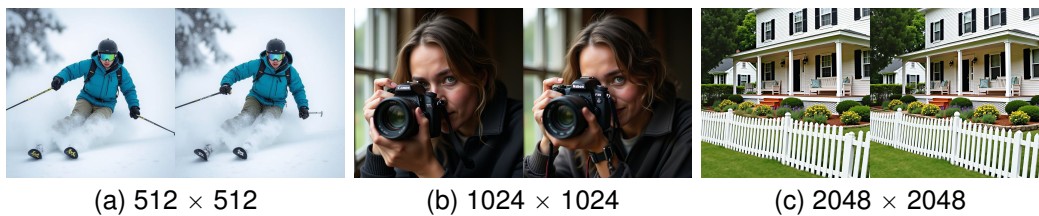

(a) $512 \times 512$            (b) $1024 \times 1024$            (c) $2048 \times 2048$

Figure 4: All images are generated by the **FLUX.1-dev** model with 25 sampling steps. The prompts used are: *"Skier gliding over fresh powder"* at resolution $512 \times 512$, *"A photographer focusing on the subject of a portrait, using natural light and composition to capture the perfect shot."* at resolution $1024 \times 1024$, and *"A charming farmhouse exterior with a white picket fence, a covered porch, and a garden full of colorful flowers and herbs."* at resolution $2048 \times 2048$. In each group, the **left image** corresponds to the **Euler** method, while the **right image** corresponds to our proposed **Euler + ParaFlow** method (parallel window size = 8, tolerance = 0.01, 0.01, and 0.0028 for the three resolutions, respectively).

**Effect of window size ($p$).** We further analyze the influence of window size $p$ on ParaFlow, using a 50-step schedule at $512 \times 512$ resolution (Table 5). The results reveal a clear trade-off: a smaller window size leads to a lower NFE and faster wall-clock time, at the cost of slightly more iterations. The window size of $p = 8$ provides the best balance, achieving a $4.3\times$ speedup with almost identical image quality to the standard Euler solver. This flexibility allows practitioners to optimize ParaFlow for their specific hardware configurations. Figure 5 visually demonstrates this effect, showing that increasing the window size preserves fidelity while offering varying levels of acceleration.

### 5.3 ADDITIONAL QUALITATIVE EVALUATION

We also provide qualitative results in Appendix D, illustrating the images generated by ParaFlow on both Stable Diffusion 3 and Flux. The samples demonstrate that ParaFlow preserves high perceptual quality and semantic alignment while significantly reducing sampling time. Together with the quantitative analyses, these results confirm ParaFlow as a general and effective solution for fast and high-fidelity diffusion sampling.

## 6 LIMITATION AND DISCUSSION

While ParaFlow achieves significant speedups, its effectiveness is predicated on the availability of parallel computing resources. The fundamental trade-off is exchanging increased computational throughput (evaluating $p$ steps in parallel) for reduced latency. This approach also increases the

| Method | Window Size | Flux Model | | | | | |
|---|---|---|---|---|---|---|---|
| | | Iters↓ | NFE↓ | CLIP↑ | FID↓ | Time (s)↓ | Speedup↑ |
| Euler | 1 | 50.00 | 50.00 | 32.13 | 52.66 | 23.80 ▬ | 1.0× |
| Euler + ParaFlow | 32 | 14.76 | 472.44 | 32.17 | 52.64 | 15.12 ▬ | 1.6× |
| | 24 | **14.75** | 354.02 | 32.17 | 52.64 | 12.21 ▪ | 1.9× |
| | 16 | 14.83 | 237.27 | 32.17 | 52.66 | 9.18 ▪ | 2.6× |
| | 8 | 15.44 | 123.51 | 32.16 | 52.58 | **5.58** ▪ | **4.3×** |

Table 5: Quantitative comparisons of different window sizes for the FLUX model over $10,000$ random samples at $512 \times 512$ with 50 sampling steps. The visual comparisons are shown in the Appendix. The best results are highlighted in **bold**. "↑" (resp. "↓") means the larger (resp. smaller), the better. Note that Euler + ParaFlow is evaluated with tolerance 0.01.

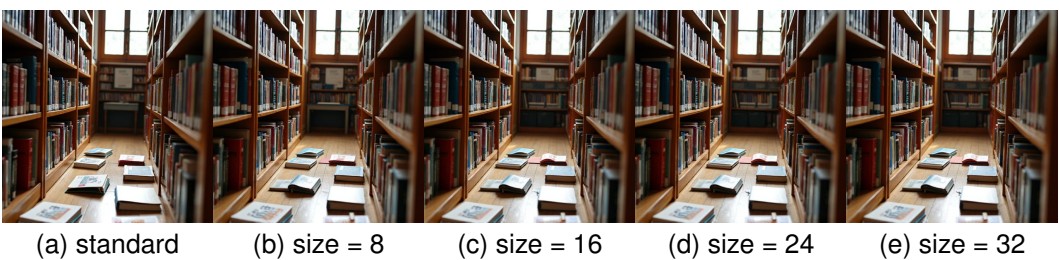

| (a) standard | (b) size = 8 | (c) size = 16 | (d) size = 24 | (e) size = 32 |
|---|---|---|---|---|

Figure 5: The above images show results from the **FLUX.1-dev** model at resolution $512 \times 512$ with 25 sampling steps. The prompt used is *"Library with lots of books on the floor"*. The leftmost image corresponds to the **Euler** method, while the other four images correspond to our proposed **Euler + ParaFlow** method under different parallel window sizes (8, 16, 24, 32) with tolerance fixed at 0.01.

memory footprint, since intermediate states within the parallel window must be stored. Our current implementation is not fully system-level optimized; techniques like kernel fusion and optimized communication collectives could further reduce overhead and amplify the speedup. Looking forward, we anticipate that ParaFlow's efficiency will scale with hardware advancements, particularly with faster inter-GPU communication and larger on-chip memory. This positions parallelized sampling as a promising direction for future large-scale generative models.

## 7 CONCLUSION

This paper introduces ParaFlow, a novel framework that reimagines the sampling process for Flow Matching models. By reformulating the inherently autoregressive ODE solving process as a system of triangular nonlinear equations, ParaFlow enables full step-level parallelism, allowing multiple sampling steps to be computed simultaneously. Our theoretical analysis confirms that this method converges to the exact trajectory of a standard sequential sampler. Empirically, on state-of-the-art models like Stable Diffusion 3 and Flux, ParaFlow delivers wall-clock speedups of up to $4.3\times$ with negligible impact on generation quality. This work opens a new avenue for accelerating generative model inference, complementing existing methods and paving the way for more efficient and interactive creative applications.

ETHICS & REPRODUCIBILITY STATEMENTS

Our work focuses on a fundamental algorithmic improvement for sampling from generative models and does not introduce new ethical concerns beyond those already associated with large-scale text-to-image models. The models used in our experiments (Stable Diffusion 3, Flux) are developed by third parties, and we use them as is. Our method could be used to accelerate the generation of harmful content, but it does not inherently make such generation easier or more likely than with standard samplers. We believe the primary positive impact is making high-fidelity generative AI more accessible for research and creative applications by lowering the inference time barrier. For reproducibility, we have detailed our methodology, algorithm, and experimental setup in the paper. We will release our source code, built upon standard open-source libraries, upon publication to allow for full verification of our results.

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

## A  USE OF LLM

During the preparation of this work, we used Large Language Models (LLMs) to assist with the writing process. The primary uses included polishing and improving the fluency of the text, generating preliminary drafts of proofs, and assisting in the creation and formatting of tables. After using these tools, the author(s) reviewed and edited the content extensively. We take full responsibility for the entire content of this publication, including the ideas, proofs, and presentations ultimately contained in the final manuscript.

## B  PROOF OF PROPOSITION 1

We first show that the triangular nonlinear equation (TNE) system in Eq. (4) has a unique root. Suppose there exist two distinct solutions $\{A_0, \cdots, A_N\}$ and $\{B_0, \cdots, B_N\}$. By construction, for any $i \in \{1, \ldots, N\}$ these solutions satisfy:

$$\begin{cases} A_i = F_{i-1}(A_0, \cdots, A_{i-1}) \\ B_i = F_{i-1}(B_0, \cdots, B_{i-1}). \end{cases} \tag{7}$$

By induction, assume $A_j = B_j$ for all $0 \leqslant j \leqslant i - 1$. Then,

$$A_i = F_{i-1}(A_0, \cdots, A_{i-1}) = F_{i-1}(B_0, \cdots, B_{i-1}) = B_i, \tag{8}$$

which implies $A_i = B_i$. Therefore, all components of the two solutions must coincide, and the root is unique.

Next, we show that this unique root coincides with the autoregressive trajectory. From Eq. (3), the autoregressive update satisfies

$$x_{t_i} = x_{t_0} + \sum_{j=0}^{i-1} h_j v(x_{t_j}, t_j, \theta). \tag{9}$$

Meanwhile, the definition of $F_{i-1}$ in Eq. (5) gives

$$\hat{x}_{t_i} = F_{i-1}(\hat{x}_{t_0}, \cdots, \hat{x}_{t_{i-1}}) = \hat{x}_{t_0} + \sum_{j=0}^{i-1} h_j v(\hat{x}_{t_j}, t_j, \theta). \tag{10}$$

By induction on $i$, since $\hat{x}_{t_0} = x_{t_0}$, it follows that $\hat{x}_{t_i} = x_{t_i}$ for all $i \in \{0, \ldots, N\}$. Thus, the TNE system admits a unique solution identical to the autoregressive trajectory.

## C  PROOF OF PROPOSITION 2

We analyze the fixed-point iteration defined in Eq. (6):

$$\hat{x}_{t_0}^{(k+1)} = x_{t_0}, \quad \hat{x}_{t_i}^{(k+1)} = F_{i-1}(\hat{x}_{t_0}^{(k)}, \cdots, \hat{x}_{t_{i-1}}^{(k)}), \quad i \in \{1, \ldots, N\}. \tag{11}$$

We prove by induction that after $k$ iterations, $\hat{x}_{t_j}^{(k)} = x_{t_j}$ for all $j \leqslant k$.

**Base case** ($k = 1$). By definition, $\hat{x}_{t_0}^{(1)} = x_{t_0}$. Moreover,

$$\hat{x}_{t_1}^{(1)} = F_0(\hat{x}_{t_0}^{(0)}) = F_0(x_{t_0}) = x_{t_1}. \tag{12}$$

Thus, indices 0 and 1 are exact after the first iteration.

**Inductive step.** Suppose $\hat{x}_{t_j}^{(k)} = x_{t_j}$ for all $j \leqslant k$. At the $(k+1)$-th iteration,

$$\hat{x}_{t_{k+1}}^{(k+1)} = F_k(\hat{x}_{t_0}^{(k)}, \cdots, \hat{x}_{t_k}^{(k)}) = F_k(x_{t_0}, \cdots, x_{t_k}) = x_{t_{k+1}}, \tag{13}$$

Hence, the $(k+1)$-th variable becomes exact, while all previously correct variables remain unchanged, since their updates depend only on values already exact.

**Conclusion.** After $N$ iterations, $\hat{x}_{t_j}^{(N)} = x_{t_j}$ for all $j = 0, \ldots, N$, establishing exact convergence in at most $N$ steps.

## D  ADDITIONAL QUALITATIVE RESULTS

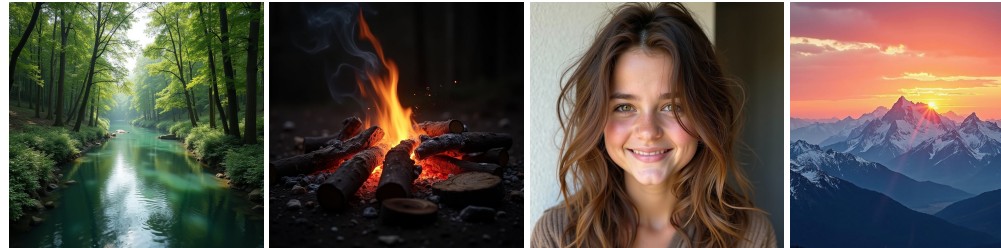

(a) A peaceful river surrounded by tall trees and lush greenery on both sides. (b) The glowing ember of a dying fire, surrounded by the charred remains of last night's logs and a few wispy strands of smoke curling upwards. (c) Women in swimwear performing synchronized swimming routine. (d) A stunning mountain range at sunset, with snow-capped peaks fading into a brilliant orange and pink sky that stretches endlessly in every direction.

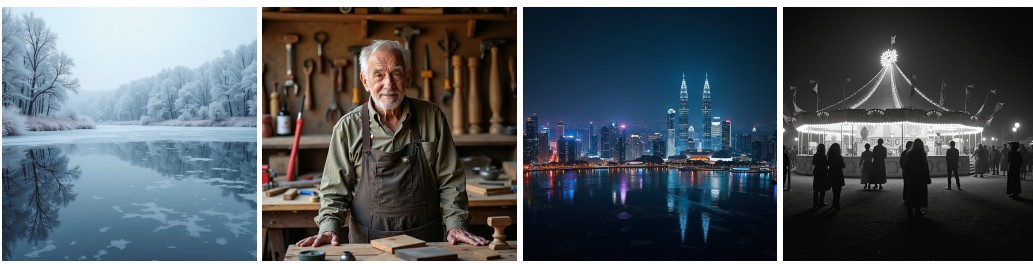

(e) Frozen lake with frosty trees reflected in the water (f) A stunning portrait of a wise old wood-worker standing in front of a wooden workbench, surrounded by tools and half-finished projects. (g) LED-lit cityscape at night (h) Old, creepy carnival at midnight

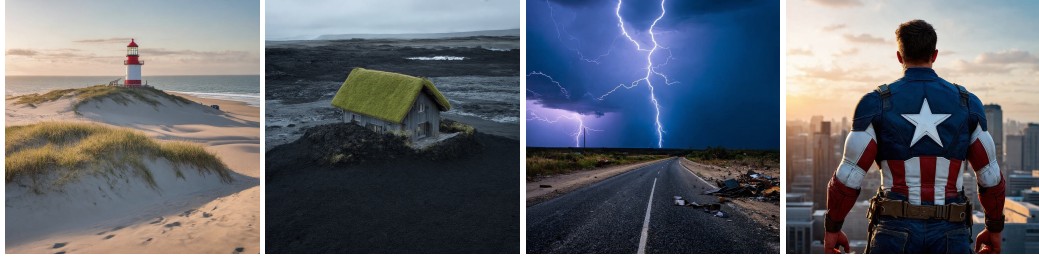

(i) Envision a serene coastal scene featuring a rustic lighthouse situated among towering sand dunes and vast expanses of sandy beach, as the sun rises over the horizon. (j) An intricately carved Icelandic turf roofed cabin standing proudly amidst a windswept landscape of black sand beaches and jagged rock formations. (k) Show a dramatic split-second moment of a thunderstorm unleashing its fury on a deserted highway, with lightning flashing across the darkening sky and debris scattered everywhere. (l) Depict Captain America standing on a rooftop, looking out over a cityscape at sunset, with his eyes narrowed in focus as he surveys his surroundings.

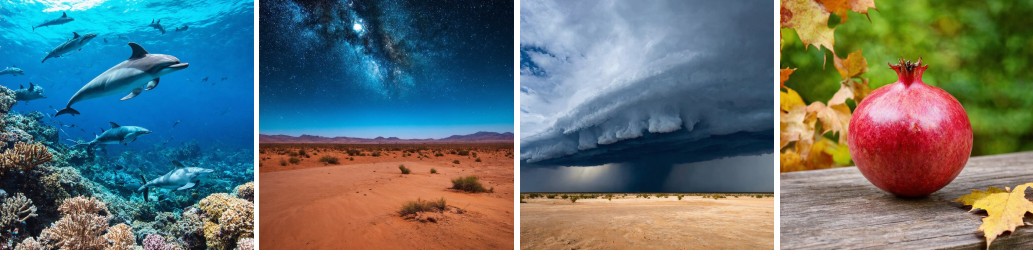

(m) A pod of dolphins swimming through coral reef (n) Desert landscape under stars (o) A massive thunderstorm unfolding over the Great Sandy Desert, with towering cumulus clouds stretching across the sky. (p) A vibrant pomegranate fruit sitting alone on a rustic wooden table, surrounded by lush greenery and autumn leaves.

Figure 6: Qualitative results generated by Euler + ParaFlow with different prompts on the FLUX.1-dev and stable-diffusion-3-medium models at a resolution of $1024 \times 1024$.

