# OpenReview forum: "ParaFlow: Parallel Sampling for Flow Matching Models"
_ICLR.cc/2026/Conference — ICLR 2026 Conference Withdrawn Submission_

### Official Review · Reviewer_yTc5 · 2025-10-21

**Soundness:** 2
**Presentation:** 3
**Contribution:** 2
**Rating:** 4
**Confidence:** 3

**Summary:**

This paper addresses the fundamental challenge of accelerating the inherently autoregressive sampling in Flow Matching (FM) models such as Stable Diffusion 3 and Flux from a numerical systems perspective. It introduces a unified framework that recasts the autoregressive sampling process as solving a system of triangular nonlinear equations (TNEs), enabling a paradigm shift toward non-autoregressive sampling with parallel vector field computation across multiple timesteps. Within this generic framework, the paper establishes two key points: (1) the TNE system has a unique solution that precisely corresponds to the autoregressive sampling trajectory; (2) solving the TNE system ensures convergence to this exact trajectory in far fewer sequential iterations. Building on these insights, this paper presents ParaFlow, a training-free, step-parallel sampler for accelerating autoregressive FM samplers. Extensive experiments validate that ParaFlow reduces sequential sampling steps by up to 4× and achieves significant wall-clock speedup of up to 4.3×, with negligible impact on FID and CLIP scores. The source code will be publicly released.

**Strengths:**

1. Clear motivation and good writing.
2. Introduces a unified framework that recasts autoregressive sampling as solving triangular nonlinear equations (TNEs), enabling a paradigm shift to non-autoregressive sampling with parallel vector field computation across multiple timesteps.
3. Presents a training-free, step-parallel sampler, avoiding extra training costs.
4. Achieves up to 4× reduction in sequential sampling steps and 4.3× wall-clock speedup, with negligible impact on FID and CLIP scores.

**Weaknesses:**

1. The experiments lack validation in other domains, e.g., text-to-video generation and class-to-image generation.

2. The paper is missing related work on deep equilibrium models [1, 2, 3, 4], which are actually trainable fixed-point iteration models.

3. The paper also omits related work on Jacobian decoding [5], which similarly involves fixed-point iterations.

4. Would a fixed-point iteration solver—such as Anderson acceleration—be helpful?

I would be willing to raise the score if these concerns are addressed.


[1] Bai, Shaojie, J. Zico Kolter, and Vladlen Koltun. "Deep equilibrium models." Advances in neural information processing systems 32 (2019).

[2] Pokle, Ashwini, Zhengyang Geng, and J. Zico Kolter. "Deep equilibrium approaches to diffusion models." Advances in Neural Information Processing Systems 35 (2022): 37975-37990.

[3] Bai, Shaojie, et al. "Deep equilibrium optical flow estimation." Proceedings of the IEEE/CVF conference on computer vision and pattern recognition. 2022.

[4] Wang, Shuai, Yao Teng, and Limin Wang. "Deep equilibrium object detection." Proceedings of the IEEE/CVF international conference on computer vision. 2023.

[5] Song, Yang, et al. "Accelerating feedforward computation via parallel nonlinear equation solving." International Conference on Machine Learning. PMLR, 2021.

[6] https://en.wikipedia.org/wiki/Anderson_acceleration

**Questions:**

see weakness

---

### Official Review · Reviewer_xZyH · 2025-10-26

**Soundness:** 2
**Presentation:** 2
**Contribution:** 1
**Rating:** 2
**Confidence:** 3

**Summary:**

This paper proposes ParaFlow, a parallel sampling algorithm for Flow Matching (FM) generative models such as Stable Diffusion 3 and Flux. Instead of sequentially integrating the learned ODE, ParaFlow reformulates the sampling process as a system of triangular nonlinear equations (TNEs), which can be solved using a parallel fixed-point iteration (FPI) scheme. This approach allows simultaneous computation of multiple ODE steps, thus reducing sequential latency. Experiments on Stable Diffusion 3 and Flux demonstrate wall-clock speedups of up to 4.3× with negligible changes in FID and CLIP scores.

**Strengths:**

1. The paper gives a clean explanation of autoregressive sampling property of ODE integration in flow matching, and the idea of equating euler iteration with TNE is novel and interesting.

2. ParaFlow can be directly applied to existing flow-based models without retraining, making it practically feasible.

3. The method achieves noticeable acceleration with minimal degradation in visual quality on strong baselines (Stable Diffusion 3, Flux).

**Weaknesses:**

1. The core formulation mainly relies on classical ODE discretization and fixed-point iteration theory. The novelty lies more in application and engineering than in new theoretical development. Both propositions 1 and 2 are common senses in ODE courses.

2. The paper does not quantitatively relate the number of parallel iterations $K$ to the actual error or convergence precision. A clear trade-off curve between accuracy and iteration count is missing.

3. The experiments are confined to only two pretrained models (Stable Diffusion 3 and Flux) and mostly show image-level metrics (FID, CLIP). There are no comparisons with other recent parallel diffusion solvers such as ParaSolver (Lu et al., 2025) or ParaTAA (Tang et al., 2024).

4. Although the method reduces sequential steps, it increases total NFE substantially. The impact on total compute cost and GPU utilization efficiency is not rigorously analyzed.

**Questions:**

Q1: Could the authors provide a quantitative relation between the number of parallel iterations $K$ and the achieved numerical accuracy?

Q2: Have the authors compared ParaFlow with other parallel ODE or diffusion solvers (e.g., ParaTAA, ParaSolver)? This would clarify whether the benefit comes from the specific TNE formulation or general step-parallel strategies.

---

### Official Review · Reviewer_nM3s · 2025-10-31

**Soundness:** 1
**Presentation:** 2
**Contribution:** 1
**Rating:** 0
**Confidence:** 5

**Summary:**

The paper proposes ParaFlow, a training-free, step-parallel sampler for Flow Matching (FM). It rewrites the discrete ODE sampling path as a triangular system of nonlinear equations and solves it via fixed-point iterations (optionally in sliding windows). The claim is that this “solves the whole trajectory simultaneously,” reducing sequential dependencies and enabling parallel evaluation across timesteps. Experiments on large text-to-image FM models report latency gains with small quality changes.

**Strengths:**

Clarity of construction: The triangular formulation and fixed-point solver are simple and easy to implement; drop-in for existing FM pipelines.

Practicality: Training-free; compatible with common samplers and amenable to batching across time.

Empirical promise: Shows meaningful wall-clock reductions under certain hardware settings; includes window/tolerance ablations.

**Weaknesses:**

1. Overstated claim: “Eliminates sequential dependencies” is misleading—sequential rounds remain as fixed-point iterations $K$. Without a proof that $K \ll N$, gains are empirical only.
2. Theory is shallow:  Convergence “in at most $N$” follows structurally from the triangular map; no FM-specific contraction/rate or tolerance-to-trajectory error bound is given.
3. Novelty vs. prior parallel sampling: Close to existing fixed-point/Picard-style parallelization in diffusion; differentiation is incremental.
4. Fairness of comparisons: Primarily Euler; lacks comparisons to stronger sequential baselines (Heun/DPM-Solver etc.) at matched quality/latency.

**Questions:**

1. Do you have *a priori* conditions (e.g., Lipschitz/step-size) guaranteeing $K=O(1)$ or $K \ll N$? If not, please soften the “eliminates dependencies” claim.
2. How does ParaFlow perform with non-Euler solvers for FM, and does triangularity still hold without extra predictors?
3. Please add fair baselines: stronger sequential solvers and schedule-optimized or distilled alternatives at matched compute/latency.

---

### Note · Authors · 2025-12-22

I have read and agree with the venue's withdrawal policy on behalf of myself and my co-authors.